# Following replicative DNA synthesis by time-resolved X-ray crystallography

Nicholas Chim [1], Roman A. Meza[1], Anh M. Trinh[1], Kefan Yang[4] & John C. Chaput [1,2,3,4 ✉]

The mechanism of DNA synthesis has been inferred from static structures, but the absence of temporal information raises longstanding questions about the order of events in one of life's most central processes. Here we follow the reaction pathway of a replicative DNA polymerase using time-resolved X-ray crystallography to elucidate the order and transition between intermediates. In contrast to the canonical model, the structural changes observed in the time-lapsed images reveal a catalytic cycle in which translocation precedes catalysis. The translocation step appears to follow a push-pull mechanism where the O-O1 loop of the finger subdomain acts as a pawl to facilitate unidirectional movement along the template with conserved tyrosine residues 714 and 719 functioning as tandem gatekeepers of DNA synthesis. The structures capture the precise order of critical events that may be a general feature of enzymatic catalysis among replicative DNA polymerases.

---

[1] Department of Pharmaceutical Sciences, University of California, Irvine, CA, USA. [2] Department of Chemistry, University of California, Irvine, CA, USA. [3] Department of Molecular Biology and Biochemistry, University of California, Irvine, CA, USA. [4] Department of Chemical and Biomolecular Engineering, University of California, Irvine, CA, USA. ✉email: jchaput@uci.edu

DNA polymerases are among the most widely studied enzymes in biology due to their importance in DNA replication and repair[1]. In each reaction cycle, a DNA primer-template duplex is extended by one nucleotide that is complementary to the templating base[2]. This reaction has been studied using kinetic[3], biochemical[4–6], and computational methods[7,8]. These data, along with crystal structures showing numerous polymerases in various pre- and post-catalytic states[9–14], have been used to infer a catalytic mechanism that involves (i) formation of the enzyme-DNA complex, (ii) binding to the incoming nucleoside triphosphate (dNTP), (iii) chemical bond formation, and (iv) translocation[15]. However, the absence of temporal information raises important questions about the order of events in one of life's most central processes.

Recently, we investigated the mechanism of a bacterial DNA polymerase I member isolated from the thermophilic species Geobacillus stearothermophilus, Bst, by X-ray crystallography (Supplementary Fig. 1)[16]. Static structures of the product of a single nucleotide addition reaction to the 3' end of a DNA primer were obtained in two different reaction environments. The first structure was obtained by crystallizing the product of a primer-extension reaction that was performed in a buffered solution, while the second structure was obtained in crystallo by soaking the dNTP substrate into a preformed crystal of the binary Bst-DNA complex. Although both structures yield the desired post-catalytic state with the primer strand extended by one nucleotide (Supplementary Fig. 2), the structures depict strikingly different active site configurations[16]. In particular, the next 5' unpaired base in the template is located in two very different positions, a trend that we found to be highly reproducible depending on the reaction environment (solution versus crystalline)[16]. In the solution-catalyzed structure, the next templating base is held outside the active site pocket through a stacking interaction with Y719 (Supplementary Fig. 2a), while the in crystallo-catalyzed structure shows the base occupying a well-defined hydrophobic pocket formed between the O and O1 helices of the finger sub-domain, as first described by Beese and colleagues (Supplementary Fig. 2b)[17]. Additionally, Y714, a critical residue for DNA synthesis[4], forms a stacking interaction with the primer strand in the solution-catalyzed structure, but shifts to the other side of the duplex in the in crystallo-catalyzed structure to form a stacking interaction with the template.

In seeking to rationalize these differences, we considered the possibility that the two structures capture different intermediates along the same reaction pathway. However, the contribution of these structures to the mechanism of DNA synthesis was ambiguous, owing to a lack of temporal information needed to understand the transition between various states in a complex reaction cycle. If we assume, based on the position of the next templating base, that the solution-catalyzed structure defines an earlier intermediate in the reaction pathway (Supplementary Fig. 2), such as the starting point of the next catalytic cycle, and that the in crystallo-catalyzed structure captures a subsequent intermediate, then additional steps would still be required for the templating base to enter the active site so that it can form a Watson-Crick base pair with the incoming dNTP substrate.

Here we follow the reaction pathway of Bst DNA polymerase by time-resolved X-ray crystallography. In contrast to the canonical model where translocation is thought to be the last step in the catalytic cycle, time-lapsed images capturing the order and transition between intermediates demonstrate that translocation precedes phosphodiester bond formation in the mechanism of DNA synthesis. The translocation step appears to follow a push-pull mechanism where the O-O1 loop of the finger subdomain acts as a latch or pawl to facilitate unidirectional movement along the template. Close examination of the active site suggests that

conserved tyrosine residues 714 and 719 functioning as tandem gatekeepers of DNA synthesis. The precise order of events suggests that the reaction mechanism may be a general feature of replicative DNA polymerases.

## Results and discussion

**Reaction optimization and data collection.** Recognizing that Bst DNA polymerase (DNAP) is functional in crystallo, we postulated that time-resolved X-ray crystallography could be used to determine the identity and order of intermediates in the catalytic cycle[18]. This method has been used previously to study the mechanism of DNA repair enzymes[19–22] and to capture the intermediate responsible for non-enzymatic RNA synthesis[23]. In order to visualize the catalytic intermediates, it was first necessary to identify conditions that allow the reaction to proceed at rates that are significantly slower than an unconstrained solution reaction. This was ultimately achieved by performing the reaction in buffer containing trace amounts of $Mg^{2+}$ ions, which allows extension of the primer by one nucleotide in minutes rather than seconds.

Crystals of Bst DNAP (299 – 876 amino acids) in complex with double-stranded (ds) DNA were grown from a primer-extension reaction that was performed in a buffered solution as described above[16]. These crystals, which depict the polymerase bound to dsDNA with a five-nucleotide overhang (3'-TGCAG-5'), were used to initiate the time-resolved study because they consistently diffract to higher resolution than equivalent crystals grown in the absence of a nucleotide addition step. For each time-dependent assay (Supplementary Fig. 3, Supplementary Table 1), individual crystals were transferred to an equilibration buffer [0.1 M MES (pH 7.0), 50% $(NH_4)_2SO_4$, and 2.5% MPD] for 30 min at 294 K. The catalytic cycle was then initiated by transferring the crystals to reaction buffer containing 2'-deoxyadenosine triphosphate (dATP) [equilibration buffer supplemented with dATP]. In total, we collected >100 datasets by stopping the reaction at various time points with liquid nitrogen and solving the structures by X-ray crystallography (Supplementary Table 2). Analysis of the resulting data indicates that all of the structures are identical except for the active site configuration. The datasets acquired at 1, 4, 8, and 120 min (refined to 1.9–2.3 Å, Supplementary Table 3) reproducibly yield unique intermediates with high occupancy (>80%).

**The initiation step of DNA synthesis.** The initiation step of DNA synthesis was investigated by monitoring extension of the primer by a nucleotide that is complementary to the first unpaired base in the template. The starting conformation observed at 0 min (Fig. 1) shows an active site configuration in which Y714 stacks directly over the 3' end of the primer (position 10, P10) and the unpaired portion of the template strand takes a sharp turn (~90°) at the base of the finger subdomain with the next 5' base (thymine at template position 5, T5) held outside the active site by a stacking interaction with the side chain of Y719. By 1 min, a small conformational change occurs at the base of the O-helix, leading to an opening of the O-O1 loop connecting the O and O1 helices in the finger subdomain (Fig. 1). This movement of the finger subdomain causes the dsDNA duplex to shift ~6 Å to a position where Y714 now stacks directly over the center of the DNA duplex. Electron density of the thymine base at position T5 is weakened, indicating that the next unpaired base in the template is transitioning to a subsequent position in the reaction cycle.

By 4 min, conformational change at the base of the finger subdomain is complete, allowing the O-O1 loop to form a well-developed hydrophobic pocket that is fully occupied by the

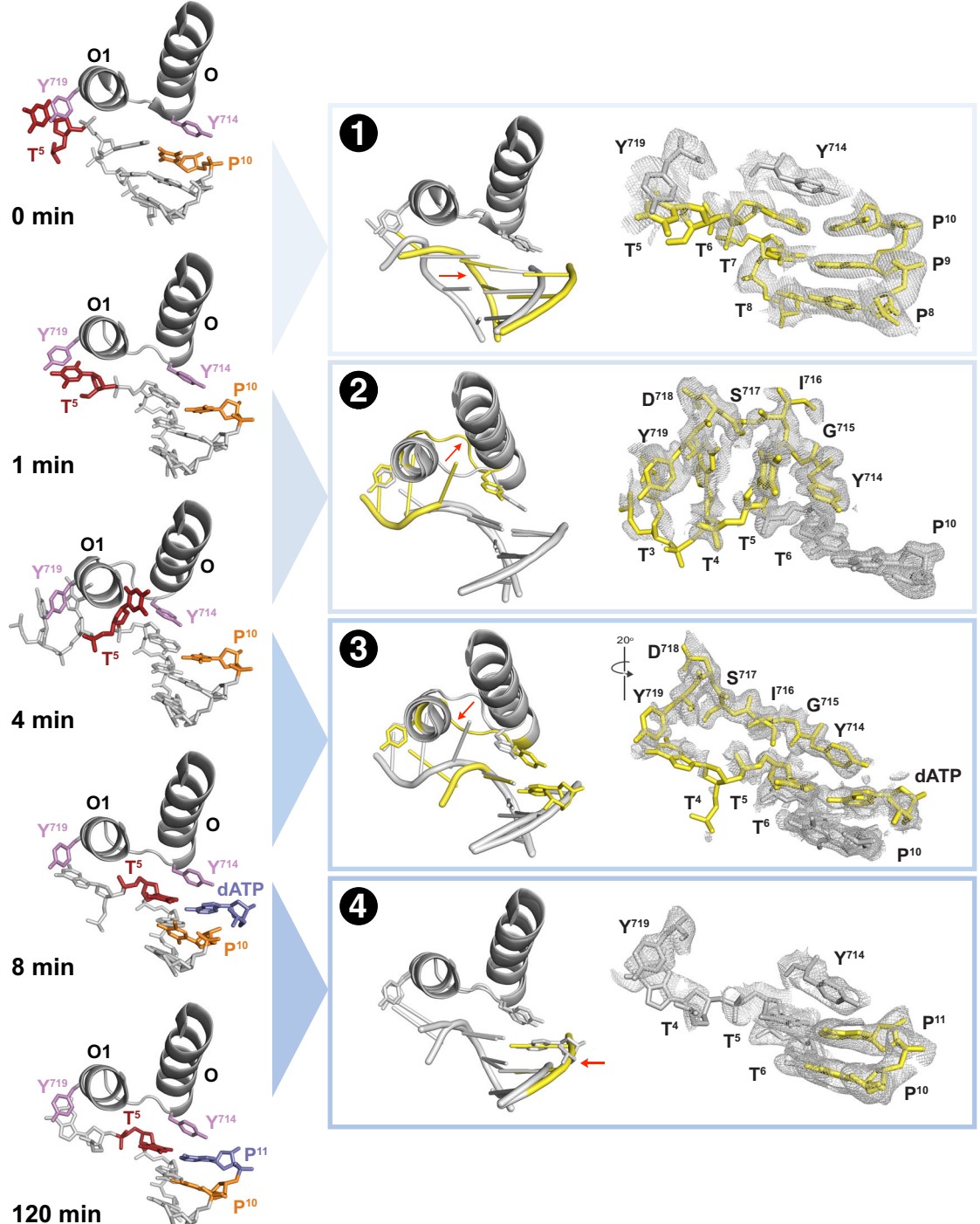

**Fig. 1 Time-ordered images capture the initiation step of DNA synthesis.** X-ray crystal structures of polymerase intermediates observed between 0–120 min (left column). Cartoon overlays and polder maps contoured at 2–4 σ (blue boxes). Red arrow indicates conformational changes between structures. The transitions are labeled: (1) movement of the DNA duplex, (2) opening of the O-O1 loop to form a hydrophobic pocket, (3) closing of the O-O1 loop and (4) chemical bond formation. Color scheme: 5′ templating base (red), Y714 and Y719 (purple), 3′ nucleotide of DNA primer (orange), dATP (blue), earlier reaction time (gray), and later reaction time (yellow). Abbreviations: O (O helix), O1 (O1 helix), T (template), P (primer), Y (tyrosine), D (aspartate), S (serine), I (isoleucine), and G (glycine).

thymine base at position T5 (Fig. 1). This intermediate in the reaction cycle is identical to the in crystallo catalyzed structure previously observed by our lab and others[16,17]. In this structure, the side chain of Y714 shifts toward the hydrophobic pocket, occupying a position where the next templating base will eventually reside to form a Watson-Crick base pair with the incoming dATP substrate.

By 8 min, a reverse conformational change occurs, closing the O-O1 loop defining the hydrophobic pocket, and pushing the 5′ untemplated thymine base into the active site where it forms a Watson-Crick base pair with the incoming dATP substrate (Fig. 1). This structure, known as the open ternary complex, positions Y714 above the center of the dsDNA duplex and relies on a stacking interaction with Y719 to prevent the next 5′

unpaired based in the template from entering the active site. As expected, the phosphate tail of the incoming dATP substrate is not visible due to a limited number of electrostatic interactions available in the open conformation[24]. Close examination of the template reading frame indicates that translocation is now complete with the unpaired thymine base having transitioned from outside the active site pocket to its final pre-catalytic position opposite the incoming dNTP substrate.

No other structural changes are observed in the active site until phosphodiester bond formation is complete by 120 min. In this structure, the 3′ hydroxyl group of the primer strand forms a chemical bond with the α-phosphate of dATP, extending the primer by one nucleotide (Fig. 1). The pyrophosphate leaving group has been displaced, but is not visible. The time delay between the open ternary complex observed at 8 min and the nucleotide addition product visualized at 120 min is consistent with the long standing belief that phosphodiester bond formation is the rate-limiting step of DNA synthesis[25]. Kinetic and structural data suggest that the rate-limiting step is deprotonation of the 3′ OH group with concomitant binding of the second (or possibly third) $Mg^{2+}$ ion[19,26,27].

Absent from the catalytic cycle is an intermediate for the closed ternary complex depicting a catalytically active form of the enzyme. However, this conformation was not expected, as it requires mutations that resolve crystal contacts observed in the open ternary structure[24]. Nevertheless, the time-lapsed images reveal a catalytic cycle for the initiation step of DNA synthesis in which translocation precedes catalysis. This observation contrasts with the canonical model of DNA synthesis in which translocation is thought to follow chemical bond formation. However, the classic model is based on static structures that lack temporal resolution, and instead rely on chemical analogs to trap the polymerase at various pre- and post-catalytic steps. By contrast, time-resolved X-ray crystallography provides an accurate view of the reaction cycle by compiling a series of snapshot images that capture the reaction pathway as it occurs inside the enzyme active site.

### The elongation step of DNA synthesis.

Next, we wished to investigate the elongation cycle of DNA synthesis. X-ray crystal structures of time-resolved intermediates were solved by transferring co-crystals of the starting Bst-DNA complex to a reaction buffer containing dATP and dCTP (Supplementary Fig. 3), which are complementary to the T5 and T4 positions of the template, respectively. Analysis of >50 datasets resulted in structures (1.6–2.3 Å, Supplementary Tables 4 and 5) that reveal the presence of unique intermediates at the 4, 25 and 48 h time points. However, the 25-hour time point is approximate because this structure required a stepwise soaking protocol.

Consistent with the initiation cycle, the first nucleotide addition product is observed at the 2-hour time point (Fig. 2). This intermediate defines the start of the elongation cycle by showing the successful addition of dATP to the 3' end of the primer. By 4 h, the finger subdomain has undergone a conformational change opening the O-O1 loop to produce a hydrophobic pocket containing the guanine base at position T4 (Fig. 2). By 25 h, the reverse conformational change is complete, closing the O-O1 loop, and pushing the guanine base into the active site where it forms a nascent Watson-Crick base pair with the incoming dCTP substrate (Fig. 2). Finally, after 48 h, chemical bond formation is complete, yielding a binary Bst-DNA complex with the primer extended by 2′-deoxycytidine (Fig. 2). In this structure, the next 5′ unpaired base is visible in the hydrophobic pocket, indicating that the enzyme is preparing for another cycle of nucleotide addition.

### Implications for the mechanism of DNA synthesis.

A combined structural view of the time-ordered initiation and elongation cycles of DNA synthesis provides evidence that translocation follows a push-pull mechanism (Fig. 3, Supplementary Movie 1) where the O-O1 loop of the finger subdomain acts as a pawl, facilitating unidirectional movement of the templating base into the enzyme active site. In this mechanism, Y714 and Y719 appear to function in tandem as gatekeepers of DNA synthesis with Y714 driving the push-pull mechanism through coordination of the incoming dNTP substrate and Y719 preventing the occurrence of frameshift mutations by controlling entrance of the templating base into the enzyme active site. Consistent with this interpretation, a significant loss of enzymatic activity is observed in a primer-extension assay when both of the tyrosine side chains are mutated to serine residues (Supplementary Fig. 4). We speculate that the energy for this process is provided by the energetics of dNTP binding, which converts binding free energy into mechanical work through conformational changes in the polymerase. Similar mechanisms have been proposed for the movement of myosin on actin filaments[28].

The time-resolved images obtained for the initiation and elongation cycles of DNA synthesis (Fig. 3) indicate that the final step of translocation occurs contemporaneously with nucleotide insertion. This observation raises the interesting question of what happens when the polymerase binds to the wrong dNTP substrate, as high fidelity DNA polymerases must rapidly sample and discriminate against incorrect nucleotides. Based on previous structural and kinetic data obtained for Bst DNAP and its homolog, the Klenow fragment of Escherichia coli DNA polymerase I, we suggest that fidelity is governed by an intermediate state, known as the ajar conformation, which allows the polymerase to preview the incoming dNTP for template complementarity[29–31]. The ajar conformation is characterized by a kink in the O-helix that leads to a partially closed polymerase conformation (Supplementary Fig. 5) in which the incoming dNTP substrate pairs with the template, but the α-phosphate occupies a suboptimal geometry for chemical bond formation[29,32]. Through a combination of hydrogen bonding and geometric selection, the ajar conformation functions as a molecular checkpoint by allowing correctly aligned dNTPs to pass through to a closed, catalytically active conformation, while pausing the reaction cycle to release mispaired nucleotides from the active site.

In summary, it has long been assumed that translocation follows chemical bond formation in the mechanism of DNA synthesis by replicative DNA polymerases. However, time-lapsed images collected here clearly show that translocation precedes phosphodiester bond formation in both the initiation and elongation cycles of DNA synthesis. While other steps, such as chemical bond formation, may warrant further study, the collection of crystallographic structures refines the mechanism of DNA synthesis by adding temporal resolution to the catalytic cycle. The clarity of the data suggests that the translocation mechanism may be a general property of replicative DNA polymerases.

### Methods

**Bst expression and purification.** DH5-α cells (New England Biolabs) harboring the pGDR11-Bst plasmid were grown aerobically at 37 °C in LB medium containing 100 μg mL$^{-1}$ ampicillin. At an $OD_{600}$ of 0.8, expression of a tagless Bst (amino acid residues 299–876) was induced with 1 mM isopropyl β-D-thiogalactoside at 18 °C for 16 h. Cells were harvested by centrifugation for 20 min at 3315 g at 4 °C and lysed in 40 mL lysis buffer (50 mM Tris-Cl pH 7.5, 1 mM EDTA, 10 mM BME, 0.1 % v/v NP-40, 0.1 % v/v Tween20, 5 mg egg hen lysozyme) by sonication. The cell lysate was centrifuged at 23,708 × g for 30 min and the clarified supernatant was heat treated for 20 min at 60 °C and centrifuged again at 23,708 g for 30 min. The supernatant was loaded onto two 5 mL HiTrap Q HP

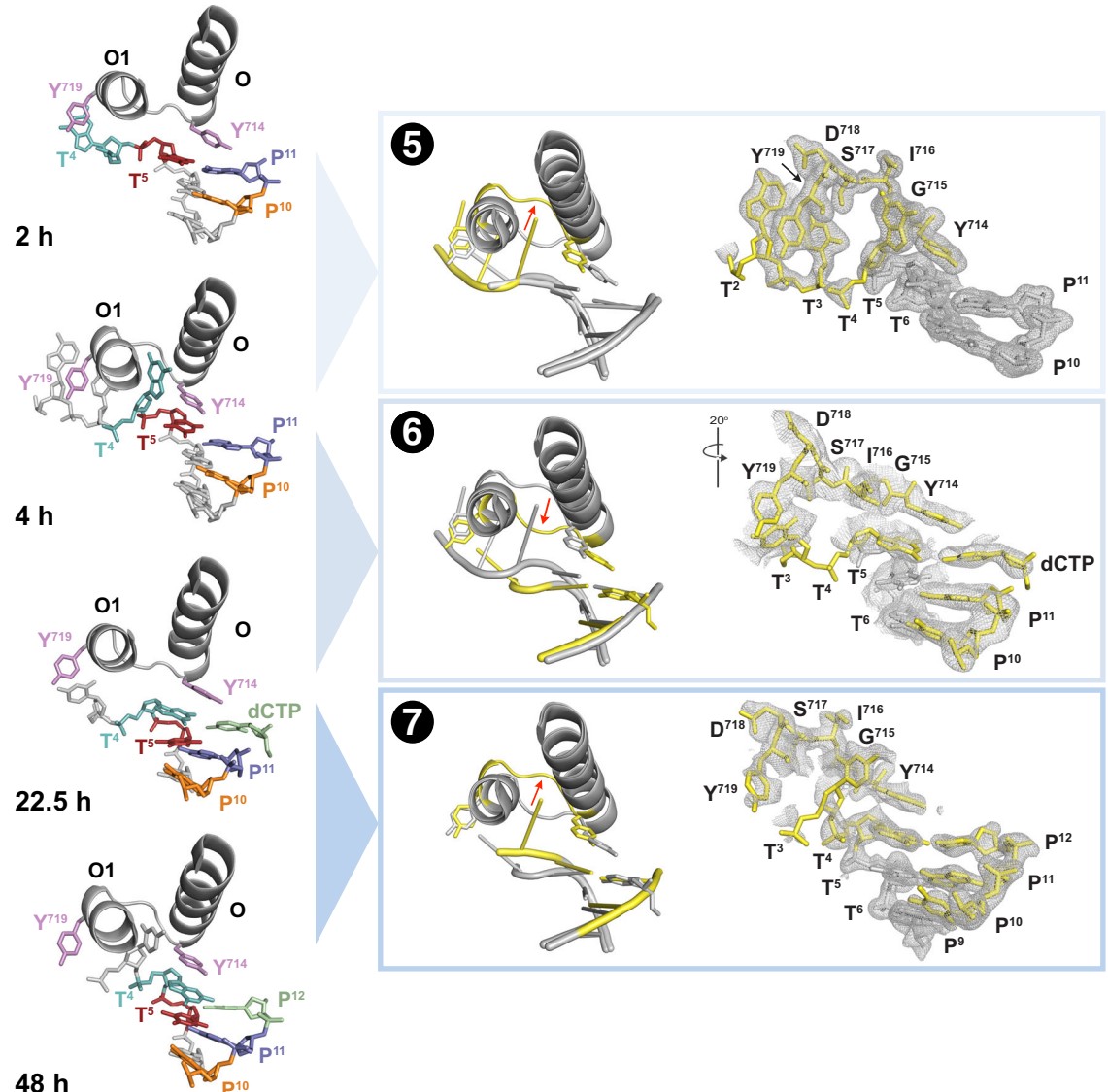

**Fig. 2 Time-lapsed images capture the elongation step of DNA synthesis.** X-ray crystal structures of polymerase intermediates observed between 2–48 h (left column). Cartoon overlays and polder maps contoured at 2–4 σ (blue boxes). Red arrow indicates conformational changes between structures. The transitions are labeled: (5) opening of the O-O1 loop to form a hydrophobic pocket, (6) closing of the O-O1 loop, and (7) reopening of the O-O1 loop following chemical bond formation. Color scheme: 5′ templating base (cyan), Y714 and Y719 (purple), 3′ nucleotide of DNA primer (blue, dCTP (green), earlier reaction time (gray), and later reaction time (yellow). Abbreviations: O (O helix), O1 (O1 helix), T (template), P (primer), Y (tyrosine), D (aspartate), S (serine), I (isoleucine), and G (glycine).

columns (GE) assembled in tandem and washed with low salt buffer (50 mM Tris-Cl pH 7.5, 100 mM NaCl, 1 mM EDTA, 10 mM BME). Bst was eluted with a high salt buffer (50 mM Tris-Cl pH 7.5, 1 M NaCl, 0.1 mM EDTA, 10 mM BME) using a linear gradient. Eluted fractions containing Bst were visualized by SDS-PAGE, pooled, and dialyzed against low salt buffer. The dialyzed sample was loaded onto a 5 mL HiTrap Heparin column (GE), washed with low salt buffer, and eluted using a linear gradient of high salt buffer. Eluted fractions containing Bst were visualized using SDS-PAGE and concentrated using a 30 kDa cutoff Amicon centrifugal filter (Millipore). Further purification was achieved by size exclusion chromatography (Superdex 200 HiLoad 16/600, GE) pre-equilibrated with Bst buffer (50 mM Tris-Cl pH 7.5, 150 mM NaCl, 1 mM EDTA, 10 mM BME). Purified Bst was concentrated to 20 mg mL$^{-1}$ for crystallization trials using a 30 kDa cutoff Amicon centrifugal filter (Millipore).

Bst mutants (Y714S, Y719S, and Y714S/Y719S) were constructed using primers (Supplementary Table 1) designed for Q5 site-directed mutagenesis (New England Biolabs) according to the manufacturer's protocol. The Bst variants were expressed in DH5-α cells as described for the wild-type polymerase. Nucleic acid contaminants present in the clarified lysate after sonication were removed by precipitation by adding 10% (v/v) polyethyleneimine (Sigma-Aldrich) to a final

concentration of 0.5%, mixing by inversion, incubating for 30 min on ice at 4 °C, and centrifuging for 30 min at 23,708 × g and 4 °C. The supernatants were transferred to fresh centrifuge tubes and the target polymerase precipitated by adding 60% (w/v) ammonium sulfate, mixing by inversion, incubating for 30 min at 4 °C, and then centrifuging for 30 min at 23,708 × g and 4 °C. Protein pellets were suspended in 4 °C low salt buffer (50 mM Tris–Cl pH 7.5, 100 mM NaCl, 1 mM EDTA, 10 mM DTT). Particulate was removed by centrifuging for 10 min at 23,708 × g and 4 °C. Bst mutants were purified by heparin affinity chromatography with step elutions of 100, 300, 500, and 1000 mM NaCl. Eluted fractions containing Bst mutants were verified by SDS-PAGE, combined, buffer exchanged to storage buffer (50 mM Tris–Cl pH 7.5, 300 mM NaCl, 1 mM EDTA, 10 mM DTT), quantified by UV absorbance at 280 nm, and stored at 4 °C.

### Crystallization procedures
*General iinformation.* All reagents purchased from commercial suppliers were of analytical grade. Stock solutions of 2-methyl-2,4-pentanediol (Hampton Research, Aliso Viejo, CA), ammonium sulfate (Teknova, Hollister, CA), 2-(*N*-morpholino) ethanesulfonic acid (Calbiochem, Burlington, MA), magnesium

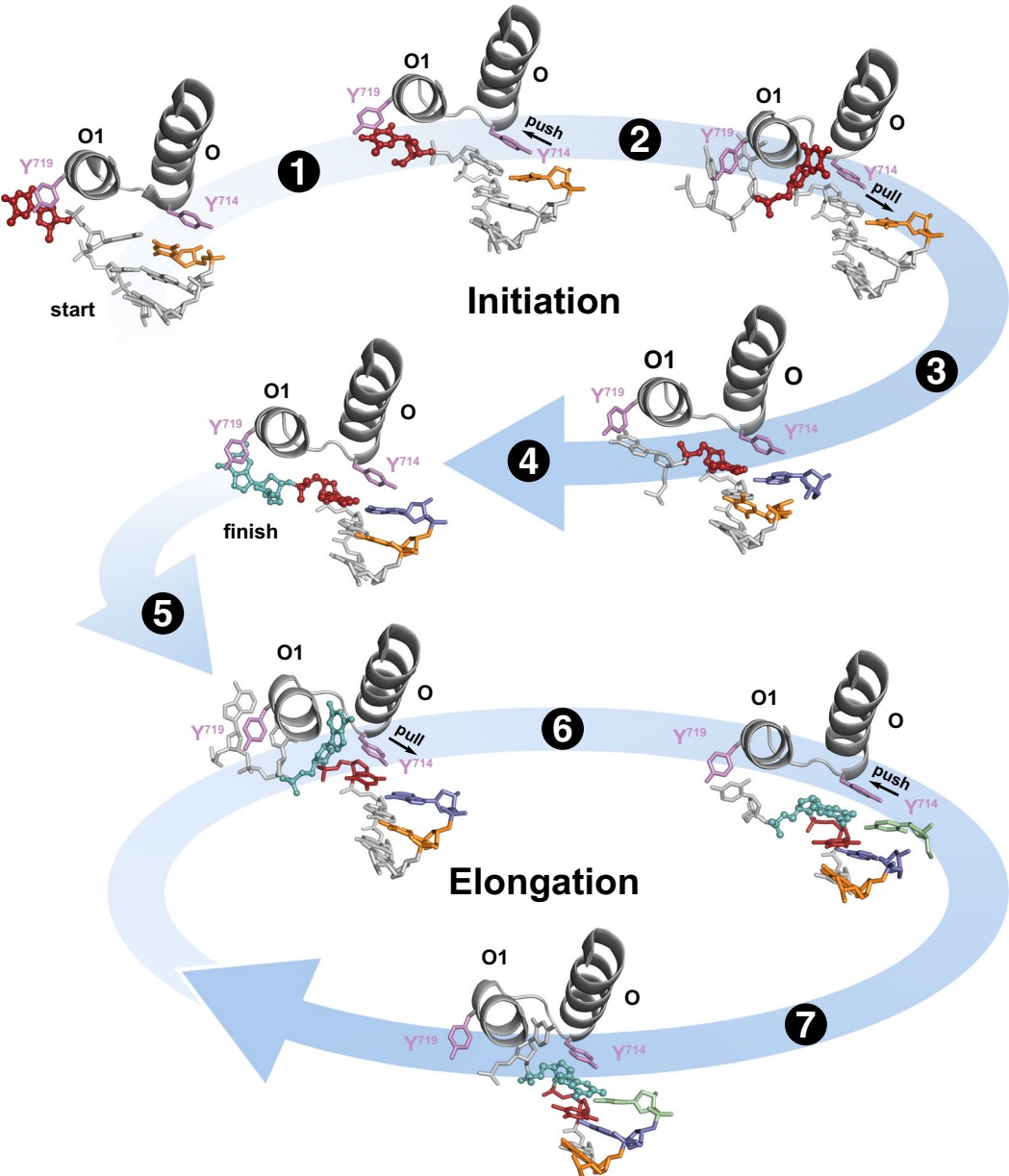

**Fig. 3 The initiation and elongation pathways of DNA synthesis.** Time-resolved images capture the precise order of intermediates in the initiation and elongation cycles of DNA synthesis. The structures imply that translocation follows a push-pull mechanism where the O-O1 loop of the finger subdomain acts as a pawl to facilitate unidirectional movement of the template through the gated actions of conserved tyrosine residues 714 and 719. The individual steps include: (1) movement of the DNA duplex, (2) opening of the O-O1 loop to form a hydrophobic pocket, (3) closing of the O-O1 loop, (4) first chemical bond formation, (5) reopening of the O-O1 loop, (6) reclosing of the O-O1 loop, and (7) second chemical bond formation with loop reopening. Color scheme: 5′ templating base (red for initiation and cyan for elongation), active site region (gray), Y714 and Y719 (purple), 3′ nucleotide on the primer (orange for initiation and green for elongation), dATP (blue), and dCTP (green). Abbreviations: O (O helix), O1 (O1 helix), and Y (tyrosine).

chloride hexahydrate (Fisher Scientific, Waltham, MA) were filtered before use during crystallization trials.

*Sample preparation.* DNA template (5′-GACGTACGTGATCGCA-3′, T) and primer (5′-GCGATCACGT-3′, P) were purchased from IDT (Coralville, IA) and were used without further purification. The P/T duplex was prepared by combining equal amounts of the primer and template strands in Bst buffer supplemented with 20 mM MgCl₂, and annealing the strands by heating for 5 min at 95 °C and cooling for 10 min on ice.

*Crystallization.* Bst, at a final concentration of 4 mg mL⁻¹, was incubated with 3 molar equivalents of the P/T duplex at 37 °C for 30 min, followed by a second incubation with 10 M excess of dTTP and 10 mM manganese chloride at 37 °C for 30 min to obtain the starting structure (0 min, 6DSY). 24–well plate hanging drop

trays were used to optimize crystal growth over a range of ammonium sulfate and MPD concentrations, with each drop containing 1 µL of sample mixed with 1 µL of mother liquor over 500 µL of mother liquor in every well. Trays were stored in the dark at room temperature and growth of Bst binary crystals was generally observed after 2 days.

*Time-resolved crystallography.* Bst binary crystals of similar size (Supplementary Fig. 1) were transferred to an equilibration buffer containing 0.1 M MES pH 7.0, 2 M (NH₄)₂SO₄, and 2.5% MPD for 30 min. The reaction was initiated by transferring crystals to an equilibration buffer supplemented with 2 mM dATP (initiation cycle experiments) or 2 mM dATP and dCTP (elongation cycle experiments). To obtain the 25.5-hour time point structure in the elongation cycle, crystals were initially soaked in dATP-supplemented equilibration buffer for 24 h before before transferring to a dCTP-supplemented equilibration buffer. After

incubation for various lengths of time, the reaction was halted by flash cooling the crystals in liquid nitrogen (Supplementary Fig. S2).

*Data collection, structure determination and refinement.* Diffraction datasets were collected at the Advanced Light Source (Lawrence Berkeley National laboratory, Berkeley, CA) from single crystals at the appropriate reaction time points. Images were indexed, integrated, and merged using XDS[33]. Data collection is summarized in Tables S2 and S4. Molecular replacement (MR) using Phaser[34] was performed using PDB structure 6DSY[16] as the search model. All final models were determined using iterative rounds of manual building through Coot[35] and refinement with phenix[36]. The final stages of refinement employed TLS parameters for all structures. The stereochemistry and geometry of all structures were validated with Molprobity[37], with the final refinement parameters summarized in Tables S3 and S5. Final coordinates and structure factors have been deposited in the Protein Data Bank. Molecular graphics were prepared with PyMOL. Polder maps for each intermediate is contoured as follows: (1) 1 min: 2.5 σ for T5–T8, 3 σ for P8–P10, Y714, and Y719; (2) 1.5 min: 2 σ for T3–T5, 2.5 σ for Y714–Y719, 4 σ for T6 and P10; (3) 8 min: 2 σ for T4 – T6 and dATP, 2.5 σ for Y714–Y719, 3 σ for P10; (4) 120 min: 2.5 σ for T4–T6, 3 σ for P10–P11, 4 σ for Y714 and Y719; (5) 4 h: 3 σ for Y714–Y719, 4 σ for T2–T6 and P10–P11; (6) 25.5 h: 2 σ for T3–T6, 2.5 σ for Y714–Y719, 4 σ for P10–P11 and dCTP; (7) 48 h: 3 σ for T3–T6 and Y714–Y719, 4 σ for P10–P12.

*Polymerase activity assay.* The poly-N templates (5′-GCN$_5$TTCGCAGTTCGCG-3′, where $N$ = A, C, G, and T) and IR680 labeled primer (5′-CGCGAACTGCGC-3′) were purchased from IDT (Coralville, Iowa). Primer extension reactions were performed for 3 min at 50 °C in ThermoPol buffer (20 mM Tris, 10 mM (NH$_4$)$_2$] SO$_4$, 10 mM KCl, 2 mM MgSO$_4$, 0.1% Triton X-100, pH 8.8), 0.5 μM primer-template duplex, 0.1 mM dNTP mix, and 3 mM MgSO$_4$ using 10 nM wild-type or mutant Bst. All reactions were quenched with 40 volumes of formamide buffer containing 25 mM EDTA. Polymerase activities were assessed by 20% denaturing urea PAGE using an Odyssey CLx imager from Li-COR (Lincoln, NE).

**Reporting summary**. Further information on research design is available in the Nature Research Reporting Summary linked to this article.

## Data availability

Coordinates and structure factors have been deposited in the PDB with the accession codes: 7K5O 7K5P 7K5Q 7K5R 7K5S 7K5T 7K5U.

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

## Acknowledgements

We would like to thank members of the Chaput lab for helpful discussions and critical reading of the manuscript. We would also like to thank A. Nikoomanzar for preparing the Bst DNA polymerase variants and M. Chaput for preparing the animated movie. This work was supported by a grant from the National Science Foundation (CHE: 2001434) to J.C.

## Author contributions

N.C. and J.C. conceived of the project and designed the experiments. N.C., R.M., A.T. and K.Y. performed the experiments. N.C. and J.C. wrote the manuscript. All authors reviewed and commented on the manuscript.

## Competing interests

The authors declare no competing interests.
