## [Peer Review File · Nature Communications]

REVIEWER COMMENTS

Reviewer #1 (Remarks to the Author):

This is a clearly written, very interesting manuscript examining the structural changes that occur during the catalytic cycle of Bst DNA polymerase (DNA Pol I). The manuscript builds off an observation that primer extension reactions carried out in the crystal versus in solution lead to different active site arrangements, particularly in regards to the templating base.

Clearly, the most interesting feature of the proposed catalytic cycle is that translocation of the templating base occurs prior to the chemical step and is proposed to be driven by the binding energy of the dNTP. I think it would be appropriate to discuss mispairing of nucleotides at this point in the discussion. In step 3 of figure 3, two important things are occurring (binding of the in-coming nucleotide and translocation). The order that the authors have assigned is reasonable, but what occurs when the wrong nucleotide is binds? I would assume it would reach the closed conformation but catalysis would not occur, then it open and release the wrong nucleotide, then another nucleotide would bind, etc. Alternatively, there could be a nucleotide "pre-check", where binding and translocation were concerted and binding of the wrong nucleotide would put you back in position 2 of figure 3. The lack of any discussion related to mispairing leaves the reader wondering why.

The biochemically assays are the weaker part of the manuscript. Besides being quite limited in scope, the lack of quantification is a problem and the results are only discussed in a generalized fashion. Are we to make any conclusions related to template sequence dependence? It seems we should, since you tested all four templating bases but it's not in the text. Is the proposed gating mechanism less important when the templating residue is a guanine? I do not think this abbreviated discussion should prevent publication because the crystallographic results are fantastic, but these assays seem like such as after-thought, it detracts from the overall paper.

Reviewer #2 (Remarks to the Author):

Nature Communication Ms-182998 has many severe crystallographic problems that need to be fixed according to PDB validation reports. Structures described by that paper do not confine the current crystallographic standards.

For example, the incoming dATP nucleotide in D_100251877 has very bad bond lengths, bond angles, torsion angles, and ring planarity. It also has a very poor RSCC of 0.67 and RSF of 0.41. The report has flagged that it is wrong! My inspection of sigmaA-weighted Fo-Fc and 2Fo-Fc maps prepared by the PDB shows that it is the most likely in an incorrect anti-glycosidic configuration!

Likewise, the incoming dCTP in D_100251880 has bad bond lengths, bond angles, torsion angles, and ring planarity. It has RMSZ = 4.90 and Z = 11. The correctness of incoming dNTPs is in question.

These errors must be fixed before this manuscript should be considered for publication anywhere. It is unclear whether the conclusion will be changed once all errors are fixed. The quality of electron density provided by the PDB validation report does not support that the authors have obtained the correct identity of nucleotides at the nascent-base pair binding pocket, not just errors in anti-syn glycosidic conformations mentioned above.

Data processing of diffraction data carried out by these authors is also problematic. The $\langle I/\sigma \rangle$ ratio in the highest 1.97-Å resolution shell of D_10025177 is 8.89. The $\langle I/\sigma \rangle$ ratio in the highest 2.00-Å resolution of D_100251881 is 9.07. The current standard is that $\langle I/\sigma \rangle$ in the highest resolution should be between 0.5 and 1, and the CC1/2 should be above 0.143. Therefore, the data processed by these authors may have systematically discarded high resolution as much as 50% of all measurable data. This procedure is called statistical gaming in X-ray crystallography, improving statistics by discarding all high-resolution data on the ground that they are poorly measured. The problem of this is present in all data sets described in this paper according to the PDB validation report.

The analysis of crystallographic data in this paper does not follow the established procedure. Given the same unit cell dimensions among these structures reported, the correct procedure is to show the

observed $F_o(\text{time point 2}) - F_o(\text{time point 1})$ isomorphous difference Fourier maps to demonstrate what structural differences between them, followed by atomic models to explain these differences. The authors have not reported any isomorphous difference between them, nor any isomorphous difference Fourier map.

Reviewer #3 (Remarks to the Author):

The manuscript by Chim et al. describes a very exciting time-resolved x-ray crystallography study of processive nucleotide incorporation by a replicative DNA polymerase. The authors collected over 100 crystal structures to capture several rounds of nucleotide incorporation and translocation during DNA synthesis of Pol I from *Geobacillus stearothermophilus* (Bst DNA polymerase).

The manuscript focuses on a description of several steps during processive synthesis (nucleotide binding, nucleotide incorporation and translocation) by Pol I. Translocation of Pol I along the template strand is facilitated by conformational changes (opening and closing) of the O-O1 helix. This helix, along with tyrosines 714 and 719, is suggested to stabilize the template strand and base in various positions to thread the template strand through the active site in a "push-pull" mechanism. While some of these observations have been described before, these new snapshots bring previous observations together into a coherent whole. The novelty appears to center around the observations regarding stabilization of the incoming template base in an extra-helical position by a tyrosine (Y719) prior to movement into a "pre-insertion" site and this is thought to influence fidelity of nucleotide incorporation. Further support of the roles of Y714 and Y719 is provided through mutagenesis into serine. The observations here nicely recapitulate, and are supported by, previous observations by the authors and work by the Beese lab ~20 years ago.

Comments:

- Did the authors attempt to solve a structure with added Mg or Mn in the time-resolved soaks? What did this structure show?
- The description together with the PDB validation reports and design of the soak experiments suggests the structures display some dynamic features, yet occupancy is >80%. Could you please show some of the $2F_o - F_c$ or $F_o - F_c$ omit maps?
- Several intermediate timepoints from various stages were collected. What did these structures show and why were they excluded? Was modelling of the partial catalysis structures attempted?
- The authors suggest bond formation or chemistry is the rate limiting step. Is there some evidence for this? Why not product release for example?
- Metals and associated active site residues appear to have been largely ignored. Is there any density for metals or pyrophosphate? Are metals bound and what happens to these or coordinating residues? Any correlation in metals or active site residues with, e.g., translocation?
- How does the conformational change in the O-O1 helix induce a 6 Å shift in DNA?
- Is there any indication as to structural reasons for the observed conformational changes and how are these substantiated in the mechanism? How are the changes in the O-O1 helix accommodated by the surrounding regions and could these accommodations influence translocation or nucleotide insertion?
- What evidence suggests dNTP binding provides the "free energy" for conformational changes of the O-O1 helix?
- How do the interactions of protein with template strand change over the course of processive incorporation?
- Is the O-O1 helix or the tyrosines 714 and 719 evolutionarily conserved? Is this push-pull mechanism relevant to other polymerases?
- What was the identity and concentration of metals during the time-resolved incubations and in the primer extension assays? Why were the assays conducted at 50 C and at pH 8.8? What would happen at 37 C and pH 7.2?
- Why did the authors mutate the tyrosines into serines? Were any other mutations made? Why does the Y719S mutation increase activity? What would the results look like on a mixed template instead of a poly-N template?
- Y714S, Y719S and the double mutant appear to be capable of processive synthesis based on Fig. 4? If so, how does the "push-pull" mechanism explain translocation, i.e., why does this still occur? How does Y719 prevent frameshift mutations? How would Y719S influence misinsertion?

Reviewer #1

1. This is a clearly written, very interesting manuscript examining the structural changes that occur during the catalytic cycle of Bst DNA polymerase (DNA Pol I). The manuscript builds off an observation that primer extension reactions carried out in the crystal versus in solution lead to different active site arrangements, particularly in regards to the templating base.

Clearly, the most interesting feature of the proposed catalytic cycle is that translocation of the templating base occurs prior to the chemical step and is proposed to be driven by the binding energy of the dNTP. I think it would be appropriate to discuss mispairing of nucleotides at this point in the discussion. In step 3 of figure 3, two important things are occurring (binding of the in-coming nucleotide and translocation). The order that the authors have assigned is reasonable, but what occurs when the wrong nucleotide binds? I would assume it would reach the closed conformation but catalysis would not occur, then it opens and releases the wrong nucleotide, then another nucleotide would bind, etc. Alternatively, there could be a nucleotide “pre-check”, where binding and translocation were concerted and binding of the wrong nucleotide would put you back in position 2 of figure 3. The lack of any discussion related to mispairing leaves the reader wondering why.

Response: We have revised the manuscript to include a discussion paragraph on the process of nucleotide selection. According to previous kinetic and structural data, the process of nucleotide selection likely occurs via the ‘ajar’ conformation, which still allows for mispairing of bases but places the α -phosphate in a suboptimal geometry for phosphodiester bonding formation. We believe that this conformation is a checkpoint that allows the polymerase to ‘pre-check’ nucleotides for shape and hydrogen bond complementarity to the templating base. We have added a new supplementary figure (supplementary Fig. 4) to illustrate the ajar conformation in comparison to the open and closed ternary structures of Bst DNA polymerase. Ongoing structural studies are now aimed at observing the process of nucleotide selection by time-resolved X-ray crystallography.

2. The biochemically assays are the weaker part of the manuscript. Besides being quite limited in scope, the lack of quantification is a problem and the results are only discussed in a generalized fashion. Are we to make any conclusions related to template sequence dependence? It seems we should, since you tested all four templating bases but it's not in the text. Is the proposed gating mechanism less important when the templating residue is a guanine? I do not think this abbreviated discussion should prevent publication because the crystallographic results are fantastic, but these assays seem like such an after-thought, it detracts from the overall paper.

Response: We agree that the biochemical assays distract from the main crystallographic findings discussed in the manuscript. Consequently, we have moved this figure to the supplementary information section. We have also revised the text describing this data to focus primarily on the loss of activity exhibited by the double-mutant polymerase, which supports the supposition that Y714 and Y719 function in tandem as gatekeepers of DNA synthesis. Although further experiments are needed to explore the biochemical implications of the gating mechanism, we feel that these studies are beyond the scope of the current manuscript.

Reviewer #2

1. Nature Communication Ms-182998 has many severe crystallographic problems that need to be fixed according to PDB validation reports. Structures described by that paper do not confine the current crystallographic standards.

Response: We understand how reviewer #2 could think that the structural violations observed for the incoming dNTP substrate are an indication of poor crystallographic standards. However, this turns-out to be an invalid concern as crystallographers familiar with replicative DNA polymerases are well aware of the fact that incoming dNTP substrates are difficult to model. This problem is due to the dynamic nature of the interaction between the incoming dNTP substrate and enzyme active site. As evidence of this problem, we offer several previously published structures of Bst DNA polymerase (PDB codes: 4YFU, 3HP6, 1LV5, 6UR9) that share this same crystallographic problem. Importantly, this error mode does not occur elsewhere in the structure, protein or DNA, and does not detract from the mechanistic conclusions of the manuscript.

2. For example, the incoming dATP nucleotide in D_100251877 has very bad bond lengths, bond angles, torsion angles, and ring planarity. It also has a very poor RSCC of 0.67 and RSF of 0.41. The report has flagged that it is wrong! My inspection of sigmaA-weighted Fo-Fc and 2Fo-Fc maps prepared by the PDB shows that it is the most likely in an incorrect anti-glycosidic configuration!

Response: We appreciate the thoroughness by which reviewer #2 evaluated the crystallographic data. However, as noted in our response to comment #1, we do not view this as valid concern, as incoming dNTP substrates are notoriously difficult to model in the enzyme active site of replicative DNA polymerases. This problem extends to the RSCC and RSF values, which describe the quality of the fit between an atomic model and its calculated electron density.

Our analyses of over 30 individual 2Fo-Fc maps (Supplementary Table 1) obtained for dATP soaks performed between 8 – 90 mins (i.e., the period of time when the dNTP substrate is present in the enzyme active site) reveal distinct, albeit weak, electron density for the dNTP substrate. Although it is difficult to refine the precise nucleotide configuration relative to the glycosidic bond, polder maps (omit maps that exclude bulk solvent around the omitted region) shown in Figure 1 unambiguously reveal the presence and position of the dNTP substrate. Importantly, carefully generated polder maps do not detect any electron density between the dNTP substrate and DNA primer, indicating that covalent bond formation has not yet occurred.

3. Likewise, the incoming dCTP in D_1000251880 has bad bond lengths, bond angles, torsion angles, and ring planarity. It has RMSZ = 4.90 and Z = 11. The correctness of incoming dNTPs is in question.

Response: This concern is related to concerns 1 and 2 discussed above. Once again, we do not view this as valid concern, as incoming dNTP substrates are notoriously difficult to model in the enzyme active site of replicative DNA polymerases.

Violations in dNTP geometry, which are related to the RMSZ and Z scores obtained for the dNTP substrate, are addressed in our response to concern #2. This is a known problem of replicative DNA polymerases that does not detract from the main conclusions of the manuscript.

4. These errors must be fixed before this manuscript should be considered for publication anywhere. It is unclear whether the conclusion will be changed once all errors are fixed. The quality of electron density provided by the PDB validation report does not support that the authors have obtained the correct identity of nucleotides at the nascent-base pair binding pocket, not just errors in anti-syn glycosidic conformations mentioned above.

Response: We do not view this as a valid concern given our responses to the related criticisms (see comments #1-3 above). In regard to the concern raised about nucleotide identity, it is clear from strong polder maps of the nucleotide addition product (Figure 1) that the incoming dNTP substrate is unambiguously incorporated as an adenosine residue onto the 3' end of the DNA primer. Thus, the time-resolved data shows the dATP entering the enzyme active site and forming a covalent bond to the DNA primer.

5. Data processing of diffraction data carried out by these authors is also problematic. The ratio in the highest 1.97-Å resolution shell of D_100025177 is 8.89. The ratio in the highest 2.00-Å resolution of D_1000251881 is 9.07. The current standard is that in the highest resolution should be between 0.5 and 1, and the CC1/2 should be above 0.143. Therefore, the data processed by these authors may have systematically discarded high resolution as much as 50% of all measurable data. This procedure is called statistical gaming in X-ray crystallography, improving statistics by discarding all high-resolution data on the ground that they are poorly measured. The problem of this is present in all data sets described in this paper according to the PDB validation report.

Response: We disagree with the assertion that the $I/\sigma(I)$ and CC1/2 values are incorrect. It is common knowledge that $I/\sigma(I)$ ratios for large macromolecules is generally >2 for the highest resolution shell while CC1/2 values are closer to 1.

Moreover, the assertion that we fiddled with our data to improve the statistical outcome of our models is false. At no time in the data processing were high-resolution data discarded to improve model statistics.

6. The analysis of crystallographic data in this paper does not follow the established procedure. Given the same unit cell dimensions among these structures reported, the correct procedure is to show the observed $F_o(\text{time point } 2) - F_o(\text{time point } 1)$ isomorphous difference Fourier maps to demonstrate what structural differences between them, followed by atomic models to explain these differences. The authors have not reported any isomorphous difference between them, nor any isomorphous difference Fourier map.

Response: We appreciate the reviewer's attention and awareness to established crystallographic procedures for analyzing time-resolved data. However, we do not feel that this is a valid concern as the normalization step, which combines and scales all datasets to a reference ground state dataset, as suggested by the reviewer was specifically developed to observe subtle differences in electron density between two closely related intermediates of an enzyme transition state that are extremely difficult to follow by X-ray crystallography. By contrast, our study examines major conformational changes that occur in the ground state as the enzyme moves through the catalytic cycle. As such, the standard approach of processing datasets individually is more than sufficient to capture all of the major conformational changes observed in the catalytic cycle. In the event that we decide to study the chemical bond forming step of the reaction, we will need to use the approach of data normalization to resolve the intermediates.

Reviewer #3

The manuscript by Chim et al. describes a very exciting time-resolved x-ray crystallography study of processive nucleotide incorporation by a replicative DNA polymerase. The authors collected over 100 crystal structures to capture several rounds of nucleotide incorporation and translocation during DNA synthesis of Pol I from *Geobacillus stearothermophilus* (Bst DNA polymerase). The manuscript focuses on a description of several steps during processive synthesis (nucleotide binding, nucleotide incorporation and translocation) by Pol I. Translocation of Pol I along the template strand is facilitated by conformational changes (opening and closing) of the O-O1 helix. This helix, along with tyrosines 714 and 719, is suggested to stabilize the template strand and base in various positions to thread the template strand through the active site in a “push-pull” mechanism. While some of these observations have been described before, these new snapshots bring previous observations together into a coherent whole. The novelty appears to center around the observations regarding stabilization of the incoming template base in an extra-helical position by a tyrosine (Y719) prior to movement into a “pre-insertion” site and this is thought to influence fidelity of nucleotide incorporation. Further support of the roles of Y714 and Y719 is provided through mutagenesis into serine. The observations here nicely recapitulate, and are supported by, previous observations by the authors and work by the Beese lab ~20 years ago.

Comments:

1. Did the authors attempt to solve a structure with added Mg or Mn in the time-resolved soaks? What did this structure show?

Response: No attempts were made at solving a structure with Mn in the time-resolved soaks. However, a few structures were solved with Mg present in the soaking buffer. These structures all showed the fully extended nucleotide addition product of the catalytic cycle, indicating that the reaction conditions are still too fast to capture intermediates in the catalytic pathway.

2. The description together with the PDB validation reports and design of the soak experiments suggests the structures display some dynamic features, yet occupancy is >80%. Could you please show some of the 2Fo-Fc or Fo-Fc omit maps?

Response: Electron density images shown on the right side of Figures 1 and 2 are polder maps, which are Fo-Fc omit maps that exclude bulk solvent in omitted region.

3. Several intermediate timepoints from various stages were collected. What did these structures show and why were they excluded? Was modelling of the partial catalysis structures attempted?

Response: The time points chosen for the catalytic cycles were empirically determined to favor the formation of discrete intermediates in the catalytic cycle. Many of the intermediate time points contained mixtures of two different conformations, indicating that the polymerase was transitioning between two different conformational states. No attempts were made to model the partial catalysis structures.

4. The authors suggest bond formation or chemistry is the rate limiting step. Is there some evidence for this? Why not product release for example?

Response: We have revised the text to include more discussion of previous kinetic and structural studies which strongly predict that the rate limiting step is either deprotonation of the 2' hydroxy group or divalent metal ion coordination. Our data corroborates these findings in a time-resolved format by showing that the slowest step in the reaction cycle is chemical bond formation. Our data also shows that product release is quite fast as the last structure in the extension cycle is already primed for another nucleotide addition step.

5. Metals and associated active site residues appear to have been largely ignored. Is there any density for metals or pyrophosphate? Are metals bound and what happens to these or coordinating residues? Any correlation in metals or active site residues with, e.g., translocation?

Response: Unfortunately, we were unable to capture the closed ternary complex of Bst DNA polymerase. However, this conformation was not expected, as it requires mutations that resolve crystal contacts observed in the open ternary structure. In the absence of the closed ternary complex, we are unable to see the coordinated divalent metal ions and corresponding amino acid side chains involved in phosphodiester bond formation. Future studies performed on a mutant Bst DNA polymerase that resolve the crystal contacts will attempt to capture the chemical bond forming step by time-resolved crystallography. However, we feel that this work is beyond the scope of the current manuscript, as it will require extensive optimization of the reaction conditions and numerous (~30-50 structures) X-ray crystallographic datasets.

6. How does the conformational change in the O-O1 helix induce a 6 Å shift in DNA?

Response: No significant conformation change (overall rmsd of 0.2) is observed in the O-O1 helix, or any other part of the protein, when the DNA duplex shifts positions in the enzyme active site. This observation was surprising to us and may suggest that DNA movement is required to properly position the template in the enzyme active site. We acknowledge that further studies are needed to understand the mechanistic rationale for this step in the catalytic cycle. However, we feel that these experiments are beyond the scope of the current manuscript.

7. Is there any indication as to structural reasons for the observed conformational changes and how are these substantiated in the mechanism? How are the changes in the O-O1 helix accommodated by the surrounding regions and could these accommodations influence translocation or nucleotide insertion?

Response: These are excellent but fundamentally difficult questions to address. Our initial analysis suggests that hundreds of small interactions are responsible for each cycle of nucleotide addition, which makes this analysis very complicated. We are currently in the process of setting up a new collaboration that will use molecular dynamics to study these interactions on all of our structural data (>100 crystal structures). This study has the potential to illuminate the hundreds of molecular interactions that drive the catalytic cycle.

8. What evidence suggests dNTP binding provides the “free energy” for conformational changes of the O-O1 helix?

Response: At present, we have no direct evidence that dNTP binding provides the free energy required for catalysis. However, such mechanisms have been proposed for the movement of myocin on actin filaments. In the context of the manuscript, this discussion is clearly offered as speculative. The actual text is as follows:

“We speculate that the energy for this process is provided by the energetics of dNTP binding, which converts binding free energy into mechanical work through conformational changes in the polymerase. Similar mechanisms have been proposed for the movement of myosin on actin filaments²⁸.”

9. How do the interactions of protein with template strand change over the course of processive incorporation?

Response: As discussed in our response to question #7 above, it's difficult to assess the contributions of all the various intermolecular interactions involved in template recognition with high confidence. As such, we are in the process of initiating a new collaboration that will explore these interactions by molecular dynamics using all of the datasets collected thus far. However, we feel that this new study is beyond the scope of the current manuscript.

10. Is the O-O1 helix or the tyrosines 714 and 719 evolutionarily conserved? Is this push-pull mechanism relevant to other polymerases?

Response: Positions 714 and 719 are evolutionarily conserved across DNA polymerase I homologs, which includes *E. coli* DNA polymerase I and *Taq* DNA polymerase. We postulate that the push-pull mechanism is relevant in these polymerases, and may be relevant to other replicative DNA polymerases. Future time-resolved X-ray crystallography studies performed on other replicative DNA polymerases will help resolve the question of generality in the mechanism of replicative DNA synthesis.

11. What was the identity and concentration of metals during the time-resolved incubations and in the primer extension assays? Why were the assays conducted at 50 C and at pH 8.8? What would happen at 37 C and pH 7.2?

Response: Crystals for the starting binary complex of Bst DNA polymerase were obtained using highly optimized crystallization conditions that contain 20 mM MgCl₂ and 10 mM MnCl₂. However, the divalent ions were depleted to trace levels by pre-soaking the crystals in an equilibration buffer for 30 mins so that the catalytic cycle can be followed by time-resolved crystallography (see reviewer #3 comment #1). By contrast, primer extension assays were performed using a commercial polymerase buffer called ThermPol, which contains 20 mM Tris, 10 mM [(NH₄)₂]SO₄, 10 mM KCl, 2 mM MgSO₄, 0.1% Triton X-100, pH 8.8. The primer extension reactions were performed at 50 C, which is the optimal temperature for Bst DNA polymerase. We have not evaluated the activity of Bst DNA polymerase at lower temperatures or at pH 7.2.

12. Why did the authors mutate the tyrosines into serines? Were any other mutations made? Why does the Y719S mutation increase activity? What would the results look like on a mixed template instead of a poly-N template?

Response: Tyrosine position 714 was extensively studied in the 1990's by Benkovic, Joyce, Kunkel, and others using the Klenow fragment of DNA polymerase I (a homolog of Bst DNA polymerase) to study fundamental questions related to the catalytic activity and mechanism of DNA synthesis. Since this is a relatively minor aspect of the paper, we decided to focus our attention on serine mutations at positions 714 and 719. No other mutations were made. We suspect that the Y719S mutation increases activity because it allows the template to pass more quickly into the enzyme active site, as it removes the stacking interaction between Y719 and the next templating base. We have not explored a mixed template sequence.

13. Y714S, Y719S and the double mutant appear to be capable of processive synthesis based on Fig. 4? If so, how does the "push-pull" -mechanism explain translocation, i.e., why does this still occur? How does Y719 prevent frameshift mutations? How would Y719S influence misinsertion?

Response: We have revised the text to include more discussion of the push-pull mechanism. We believe that Y719 prevents frameshift mutations by controlling entrance of the templating base into the enzyme active site via base stacking. Likewise, we speculate that Y714 controls fidelity through coordination (base stacking) with the incoming dNTP substrate. The reduced DNA synthesis activity observed for the double-mutant polymerase is consistent with the coordinated activity of Y714 and Y719. The continued DNA synthesis activity observed for the double mutant polymerase is likely due to the fact that catalysis is driven by a multitude of intermolecular interactions that are still largely unknown. However, we do believe that these interactions could be revealed, at least in part, from a detailed molecular dynamics study, as discussed above.

REVIEWERS' COMMENTS

Reviewer #1 (Remarks to the Author):

I am satisfied with the authors revision and appreciate the additional discussion regarding nucleotide selection. De-emphasizing the kinetic results strengthens the overall readability of the manuscript.

Reviewer #3 (Remarks to the Author):

Please know that I looked at the revision carefully. The authors addressed all concerns in the revision, and I recommend early publication.

Reviewer #1:

I am satisfied with the authors revision and appreciate the additional discussion regarding nucleotide selection. De-emphasizing the kinetic results strengthens the overall readability of the manuscript.

Response: We thank the reviewer for their positive assessment of the work.

Reviewer #3:

Please know that I looked at the revision carefully. The authors addressed all concerns in the revision, and I recommend early publication.

Response: We thank the reviewer for their positive assessment of the work.